# The research explores the predictive capacity of the shear strength of reinforced concrete walls with different cross-sectional shapes using the XGBoost model

**Hoa Thi Trinh, Tuan Anh Pham** [ID] **\*, Vu Dinh Tho, Duy Hung Nguyen**

University of Transport Technology, Hanoi, Vietnam

\* anhpt@utt.edu.vn

## Abstract

Structurally, the lateral load-bearing capacity mainly depends on reinforced concrete (RC) walls. Determination of flexural strength and shear strength is mandatory when designing reinforced concrete walls. Typically, these strengths are determined through theoretical formulas and verified experimentally. However, theoretical formulas often have large errors and testing is costly and time-consuming. Therefore, this study exploits machine learning techniques, specifically the hybrid XGBoost model combined with optimization algorithms, to predict the shear strength of RC walls based on model training from available experimental results. The study used the largest database of RC walls to date, consisting of 1057 samples with various cross-sectional shapes. Bayesian optimization (BO) algorithms, including BO—Gaussian Process, BO—Random Forest, and Random Search methods, were used to refine the XGBoost model architecture. The results show that Gaussian Process emerged as the most efficient solution compared to other optimization algorithms, providing the lowest Mean Square Error and achieving a prediction $R^2$ of 0.998 for the training set, 0.972 for the validation set and 0.984 for the test set, while BO—Random Forest and Random Search performed as well on the training and test sets as Gaussian Process but significantly worse on the validation set, specifically $R^2$ on the validation set of BO—Random Forest and Random Search were 0.970 and 0.969 respectively over the entire dataset including all cross-sectional shapes of the RC wall. SHAP (Shapley Additive Explanations) technique was used to clarify the predictive ability of the model and the importance of input variables. Furthermore, the performance of the model was validated through comparative analysis with benchmark models and current standards. Notably, the coefficient of variation (COV %) of the XGBoost model is 13.27%, while traditional models often have COV % exceeding 50%.

**Data Availability Statement:** All relevant data are within the manuscript and its Supporting information files.

## 1. Introduction

Reinforced concrete walls are important structural components, playing the role of supporting horizontal loads in high-rise buildings due to their large horizontal stiffness. To ensure bearing

**Funding:** The author(s) received no specific funding for this work.

**Competing interests:** The authors have declared that no competing interests exist.

capacity, RC walls are often designed focusing on "strong shear and weak bending". Therefore, accurate prediction of damage modes, including lateral load capacity and deformation capacity of reinforced concrete walls, is necessary during the structural design process, especially in high-rise buildings, where seismic safety is of utmost importance. Although various design codes use plane section assumptions to calculate flexural strength, the calculated values are generally accurate and consistent across different codes [1–3]. Specifically, the bending and shear resistance of load-bearing walls are determined according to current construction standards such as ACI 318–19 and EC-2. The mechanism of flexural capacity has been thoroughly explained by flexure theory [4], while the shear provisions in the ACI code are relatively straightforward [5]. Previous studies has shown that the provisions in ACI 318–19 have a low safety factor and do not take into account high-strength concrete shear walls, while the provisions in Eurocode are too conservative [6, 7].

In addition, some modern theories are also used to determine the shear strength of reinforced concrete walls such as: Modified Compression Field Theory (MCFT) [8], the Cyclic Softened Membrane Model (CSMM) [9], and the Strut-and-Tie Model (STM) [2, 10]. Although these models provide reliable estimates, they often require the establishment of Finite Element Method (FEM) or extensive theoretical calculations, resulting in low computational performance.

Recently, Machine Learning (ML) based models have demonstrated their effectiveness in forecasting the shear strength of various structural elements such as beams [11–14] and concrete columns [15]. The use of data-driven models is quite appealing due to their relative simplicity and ease of development compared to traditional models based on specific rules or hypotheses [16]. Furthermore, AI models help users reduce the burden of performing complex computational tasks. However, developing an accurate AI model also poses its own challenges. These challenges include optimizing the hyperparameters of the AI algorithm, accurately analyzing the role of input variables in predicting wall shear strength, ensuring the stability of the machine learning model, and validating the reliability of the collected data set. The purpose is to provide a simple and convenient parametric model for users to apply when designing detailed structural components. Recent studies have focused on using Machine Learning (ML) models to predict the load-bearing capacity of reinforced concrete (RC) walls, and significant findings have been published. According to Zhang et al. [17], The XBG and GB models are among the most effective models in identifying damage modes of RC walls. Specifically, the accuracy in predicting the type of damage of RC walls reaches 97%. Feng et al.[18] developed an Extreme Gradient Boosting (XGBoost) algorithm to estimate the shear strength of squat RC walls. The results demonstrated that the XGBoost model has great potential for reliably predicting shear strength, with an average prediction-to-test ratio of 1.0. Barkhordari et al. [19] also uses some Deep Neural Network models to predict the failure mode of RC walls. The research results show that the weighted average ensemble deep neural network model most accurately predicts the failure mode of RC walls, with an accuracy reaching over 0.9. Gondia et al. [20], used a genetic program to predict the shear strength of flanged squat RC walls, with a dataset of 254 samples. The results of the study revealed an explicit formula for shear strength using several mechanically guided derivatives, achieving high accuracy and demonstrating good practical applicability of the model. Keshtegar et al. [21] developed a new hybrid machine-learning model to predict the lateral strength of RC walls. Their results indicated that the hybrid artificial intelligence model, developed using an artificial neural network (ANN) optimized with an adaptive harmonic search (AHS) algorithm, achieved Outstanding performance in predicting the lateral strength of RC walls, with an average prediction-to-experiment ratio of 1.0. The combination of support vector regression (SVR) and response surface methodology (RSM) provided reasonable predictions of the lateral strength of RC walls, with an

average prediction-to-experiment ratio of 0.98 [22, 23]. Mangalathu et al. [24] investigated safety margins in RC shear walls and the lack of models for rapid failure identification. They used data from 393 shear wall experiments with varied geometries to develop prediction models. Eight machine learning methods were employed, including Naive Bayes, K-Nearest Neighbors, Decision Tree, Random Forest, AdaBoost, XGBoost, LightGBM, and CatBoost. Random Forest achieved 86% accuracy in failure mode prediction. Critical parameters influencing failure mode included wall aspect ratio, boundary element reinforcement, and wall length-to-thickness ratio. They proposed an open-source data-driven classification model for potential design applications. Barkhordari et al. [25] presented a new hybrid model based on ANN and State-of-the-art population-based algorithms to predict the Shear strength of Squat Reinforced Concrete (SRC). This study uses data from 434 experimental specimens, of only SRC wall type to train and test the ML model and the results show that the hybrid ML model can achieve high performance in calculating the shear strength of SRC wall.

From the analysis of the results obtained by applying machine learning (ML) to predict the performance of RC walls, it is clear that ML models can predict both the failure mode (classification algorithm) and the shear strength (regression algorithm) of RC walls. However, there are some limitations, especially in predicting the shear strength of RC walls (regression algorithm). These limitations include the small number of trained models, usually 1–2 models, and the lack of diversity and generalizability in the collected data, such as considering only one cross-sectional shape or focusing on a specific type of RC wall such as low RC walls, or using only one parameter optimization algorithm for the machine learning model, without any comparison to confirm the reliability of the selected optimal parameter set. Furthermore, the model architecture is often searched manually, which does not ensure the identification of models with good parameters for RC wall research. Finally, current studies mainly evaluate the sensitivity of input variables to model performance without specific analysis of the impact of changing the values of input variables on the shear strength of RC walls.

## 2. Research significance

To address the limitations of previous studies, this paper proposes the following approach to apply the ML model in predicting the shear strength of reinforced concrete walls:

1. The large dataset of RC walls collected and processed includes 1057 samples with three different cross-sectional shapes.

2. A detailed study was performed on the XGBoost model, with the parameter sets for the XGBoost model determined through three different optimization methods.

3. The role of input variables is evaluated using SHAP values for the XGBoost model, providing an explanation of the model's predictive ability.

4. The prediction ability of the XGBoost model is compared with standard design codes and existing benchmark models.

By implementing this method, the study is expected to overcome the limitations of previous studies and provide a more effective method to evaluate the shear capacity of RC walls.

## 3. Methodology

### 3.1. Data description

**3.1.1. Distribution of cross-section types.**  Reliable data is always the most important issue for machine learning models. In this study, a dataset of 1057 RC wall test samples was

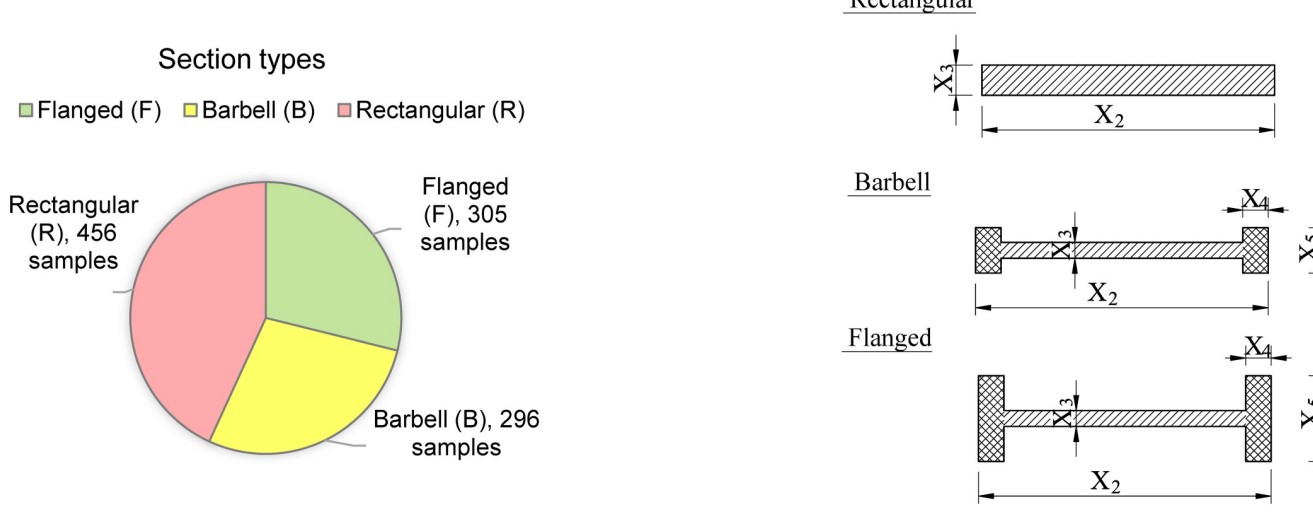

**Fig 1. Distribution of cross-section stypes.**

collected from previous literature. Data is collected from specific, reliable sources, including: 252 samples from Massone and Melo [26], 182 samples are from Ning and Li. [27], 22 samples from Sato et al. [28], a sample from Park et al. [29], 4 samples from Teng and Chandra [30], a sample from Wolschlag et al. [31], 21 samples belonged to Wang et al. [32], 4 samples belong to Vallenas et al. [33], 129 samples from Hirosawa et al. [34], 7 specimens from Barda et al. [35], 16 test results from Antebi et al. [36] and 418 samples were collected and processed from an existing database [37]. There are a total of 1057 reinforced concrete walls in the data set, in which the ratio between cross-section types is quite uniform, specifically including 456 walls with rectangular cross-sections (Rectangular), 296 walls have barbell-shaped cross-sections (Barbell) and 305 walls have flange cross-sections (Flanged). These informations are shown in Fig 1.

**3.1.2. Data statistics.** Data statistics for the model are very important. From there, the scope of application of the research will be better understood, at the same time, data statistics also clearly show the general distribution of data, in order to evaluate the balance and reliability of the training results. The RC wall tests in this database include four groups of input variables, which are geometric dimensions, reinforcement layout, material properties, and applied loads. Specifically, the detailed input characteristics are wall height (symbol is $X_1$, mm), wall length (symbol is $X_2$, mm), web thickness (symbol is $X_3$, mm), flange thickness (symbol is $X_4$, mm), flange length (symbol is $X_5$, mm), concrete compressive strength (symbol is $X_6$, MPa), web reinforcement content in the vertical direction (symbol is $X_7$, %) and the yield strength of the longitudinal web steel (symbol is $X_8$, MPa), the web reinforcement content in the horizontal direction (symbol is $X_9$, %) and the yield strength of the web longitudinal steel (symbol is $X_{10}$, MPa), the longitudinal reinforcement content (symbol is $X_{11}$, %) and the yield strength (symbol is $X_{12}$, MPa), and finally the same is the axial load (symbol is $X_{13}$, kN). The output is simply the shear strength of the wall (symbol is Y, kN). Descriptions and statistical properties of the variables are given in Table 1 (Examples of S1 Table). All input data are normalized to the range [0–1] to ensure features have equal importance in the machine learning model.

The important thing to note in the input data is that, for a rectangular cross-section, $X_4 = 0$ and $X_3 = X_5$, while the cross-section of the Barbell cross-section has a ratio $X_5/X_4 \leq 3$, the Flange cross-section has a ratio $X_5/X_4 > 3$.

**Table 1. Features of shear strength database for RC walls.**

| Features | Unit | Min | Max | Mean | Standard deviation |
|---|---|---|---|---|---|
| $X_1$ | mm | 145 | 6401 | 1274.79 | 1022.53 |
| $X_2$ | mm | 254 | 3960 | 1319.46 | 755.07 |
| $X_3$ | mm | 10 | 360 | 97.57 | 62.29 |
| $X_4$ | mm | 0 | 360 | 59.78 | 73.65 |
| $X_5$ | mm | 30 | 3045 | 253.69 | 334.05 |
| $X_6$ | MPa | 10 | 130.8 | 31.29 | 17.04 |
| $X_7$ | % | 0 | 6.24 | 0.8 | 0.72 |
| $X_8$ | MPa | 0 | 792 | 395.79 | 115.53 |
| $X_9$ | % | 0 | 3.67 | 0.67 | 0.51 |
| $X_{10}$ | MPa | 0 | 792 | 396.57 | 117.57 |
| $X_{11}$ | % | 0 | 10.58 | 3.17 | 2.01 |
| $X_{12}$ | MPa | 208.9 | 980 | 443.48 | 140.63 |
| $X_{13}$ | kN | 0 | 2429 | 293.86 | 464.31 |
| Y | kN | 15.35 | 3138.10 | 563.72 | 639.21 |

Fig 2 shows the correlation matrix of the data set, which includes 13 input variables and 1 output variable. The matrix displays the correlation coefficients between each pair of variables, where a correlation value of 1 represents a perfect positive correlation, -1 represents a perfect negative correlation, and 0 represents no correlation. The correlation matrix helps us understand the relationship between different variables and how they relate to each other. Initial analysis shows that there are both positive and negative correlations between variables and that pairs of highly correlated attributes are more interdependent. Specifically, the highest correlation coefficient is 0.89 between the two variables $X_8$ and $X_{10}$, demonstrating a close relationship between these characteristics. Additionally, geometrical parameters and loads applied to the wall have the highest correlation with output performance. Understanding the correlation matrix can help determine which features are important to the resulting characterization and which features are redundant, useful for further analysis and modeling.

## 3.2. Machine learning approaches

**3.2.1. Extreme gradient boosting machine learning model (XGBoost model).** In this study, a supervised machine learning model called eXtreme Gradient Boosting (XGB) was used to determine the shear strength of reinforced concrete walls. This is one of the most powerful and popular machine learning methods, especially in prediction and classification problems. XGBoost focuses on building a sequence of weighted decision trees, also known as boosted trees, in a gradient-boosting manner. It combines multiple single decision trees to create a powerful prediction model. However, like most other decision tree-based models, the XGB model does not have the ability to extrapolate predictions, meaning the model only predicts accurately within the range of input variables used to train the model. The general formula of the XGB model is written as follows [38]:

$$f(x) = \sum_{i=1}^{k} \gamma_i . h_i(x) \tag{1}$$

Where f(x) is output model; $\gamma_i$ is the learning rate and $h_i(x)$ is the simple tree of $i^{th}$ iteration and k is the number of iterations.

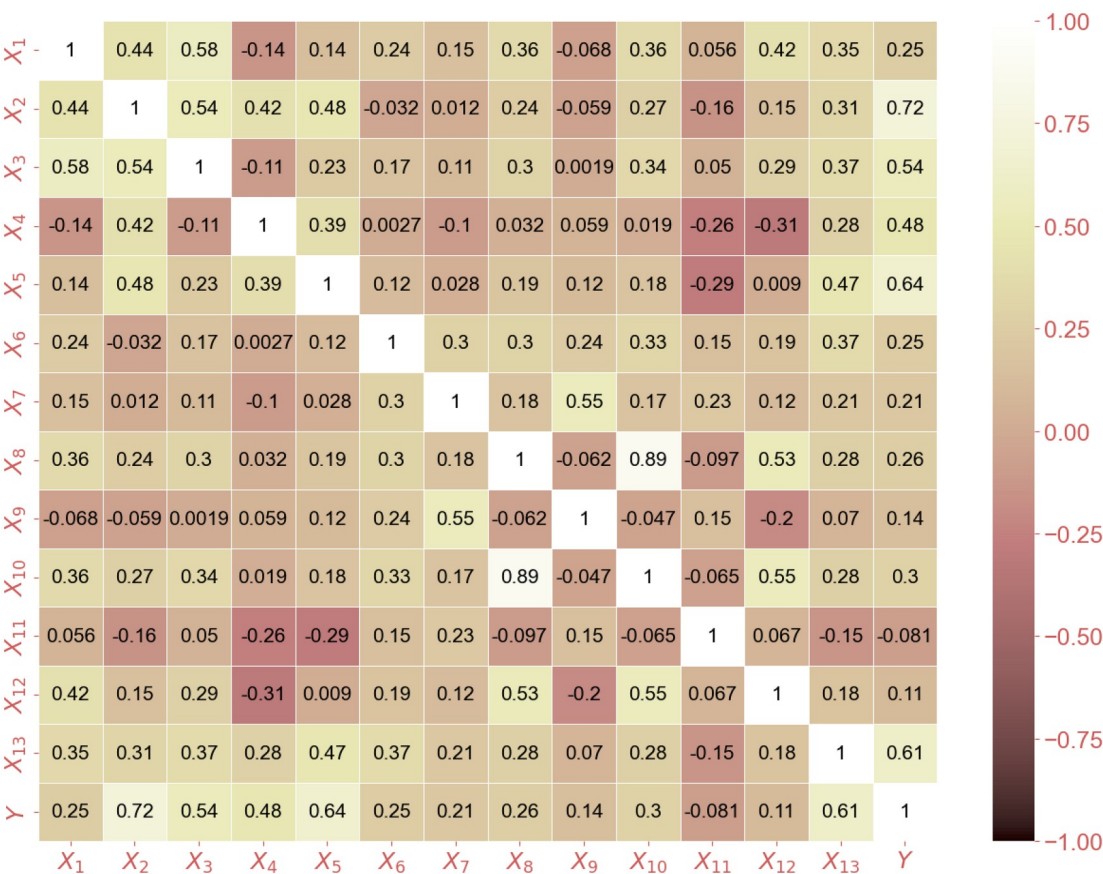

**Fig 2. Correlation matrix of the features with data 1057samples.**

**3.2.2. Optimization algorithms for machine learning models.**   To improve the performance of machine learning models, robust optimization algorithms are proposed. There are many types of optimization algorithms used to solve problems. Examples include gradient-based algorithms, grid search algorithms, stochastic search, and discrete optimization such as evolutionary algorithms or particle swarms. In this study, two typical optimization algorithms are used: the Random Search algorithm [39] and the Bayesian algorithm [40]. In the Random Search algorithm, the model's hyperparameter set is randomly selected within the search range in each iteration. This causes the algorithm to often find hyperparameter combinations better than a similar method, Grid Search. Meanwhile, Bayesian optimization differs from Random Search and Grid Search in that it takes into account past performance, while the other two methods do not take this into account. The core idea of Bayesian optimization is to build a probabilistic model of the objective function and use this model to select the most promising points for evaluation. In that sense, Bayesian Optimization first finds a set of random parameters and then evaluates the performance of this set of parameters. In the next step, the method will try to change one of the parameters and compare the model performance to see if there is any improvement. This method is especially useful in problems where the objective function is discontinuous, has no derivative, or is noisy.

**3.2.3. Performance indices of the model.**   To evaluate the performance of the established models, statistical parameters, including Correlation coefficient ($R^2$) [41], Root Mean Square Error (RMSE) [42], and mean absolute error (MAE) [42], were used. Accordingly, RMSE

evaluates the average error between the model's prediction results and the experimental results, then the smaller the RMSE value, the more accurate the prediction model. Meanwhile, $R^2$ ranges from $-\infty$ to 1, indicating a correlation between the actual value and the predicted value, meaning the higher the $R^2$ value, the better the model. The formulas of the parameters are presented below:

Coefficient of determination ($R^2$):

$$R^2 = 1 - \frac{\sum_{j=1}^{N} (y_j - y_{t,j})^2}{\sum_{j=1}^{N} (y_j - \bar{y})^2} \tag{2}$$

Root Mean Square Error (RMSE):

$$RMSE = \sqrt{\frac{1}{N} \sum_{j=1}^{N} (y_j - y_{t,j})^2} \tag{3}$$

Mean Absolute Error (MAE):

$$MAE = \frac{1}{N} \sum_{j=1}^{N} |y_j - y_{t,j}| \tag{4}$$

where $y_j$ is the actual shear strength of the $j^{th}$ sample in the dataset; $y_{t,j}$ is the predicted shear strength of the $j^{th}$ sample obtained from the ML model; $\bar{y}$ is the mean value of the actual shear strength of the data set; N is the total number of samples in the dataset.

## 4. Model implementation and prediction performance

In this study, Machine Learning models are developed based on the Python Scikit-Learn library [43]. The entire data set is randomly divided into a training set that accounts for 80% of the data and a testing set that accounts for 20% of the data. The training set is used to train and fine-tune the prediction models, while the test set is used to evaluate the performance of the models. One thing to note is that model hyperparameter tuning is performed using the K-Folds cross-validation technique on the training set. This technique is intended to ensure highly generalizable results when all data will appear in the training and validation sections respectively. In this study, K = 10 is chosen, meaning that for each hyperparameter set, the training data is divided into 10 subsets, 9 are used for training and the remaining 1 is used for validation. This will be repeated 10 times and the model's performance results will be averaged over those 10 times.

### 4.1. Hyperparameter optimization of the XGBoost model

To find the best machine learning model, three optimization solutions are used to automatically select the best set of hyperparameters for the XGBoost model: Bayesian with Gaussian process (BO-GP), Bayesian with Forest-based. Random (BO-RF) and Random Search (RS). There are 8 main hyperparameters of the model selected for optimization according to [6]. The optimization process will stop after the algorithm has performed at least 100 iterations, without the optimal result changing.

The values and hyperparameters for the XGBoost model were optimized within the specified range using BO-GP, BO-RF, and RS methods. The optimal hyperparameters, along with

Table 2. Hyperparameters for the XGBoost model.

| Hyperparameter | Meaning | Range of values | Optimal results | | |
|---|---|---|---|---|---|
| | | | (BO-GP) | (BO-RF) | (RS) |
| 'n_estimators' | Number of trees | 100–1000 | 1000 | 912 | 823 |
| 'max_depth' | Maximum depth of each tree | 3–9 | 3 | 3 | 3 |
| 'learning_rate' | Learning rate of stages | 0.05–0.30 | 0.1399 | 0.1194 | 0.109 |
| 'booster' | Booster method | 'gbtree', 'dart' | 'dart' | 'gbtree' | 'dart' |
| 'gamma' | The minimum loss to create a tree's nodes | 0.01–0.50 | 0.5 | 0.485 | 0.3077 |
| 'subsample' | The subsampling ratio in the training set | 0.60–0.90 | 0.6 | 0.697 | 0.727 |
| 'colsample_bytree' | Specifies the proportion of columns to be subsampled | 0.60–0.90 | 0.9 | 0.747 | 0.799 |
| 'reg_lambda' | Weights used in L2 regularization | 1–50 | 22 | 3 | 8 |

The optimal results show that all methods select a fairly large number of trees (from 823–1000 trees). The maximum depth of all trees is only 3 while the remaining hyperparameters are chosen differently depending on each algorithm.

the allowable value ranges, are shown in Table 2 (S1 Data). It is important to note that to avoid overfitting during training and optimization, two techniques have been applied: (1) Subsample and (2) K-fold CV. In which, the Subsample technique uses a certain proportion of input variables during training, which helps create simpler trees and avoid overfitting. Meanwhile, the K-Fold technique is used on the training data set itself, allowing the model to be trained/validated during the optimization process, all single data fold in the training set is in turn fed into training/validation, leading to training results that avoid overfitting. The hyperparameter optimization process is shown in Fig 3. It can be seen that the optimization algorithms BO-GP and BO-RF achieve convergence in about 350 iterations. Therefore, the number of iterations of the RS algorithm is also chosen to be 350 for objective comparison.

## 4.2. Evaluation of hyperparameter optimization performance

The impact of optimized hyperparameters on the performance of machine learning models is evaluated by comparing their performance with default parameters. The aggregated results for

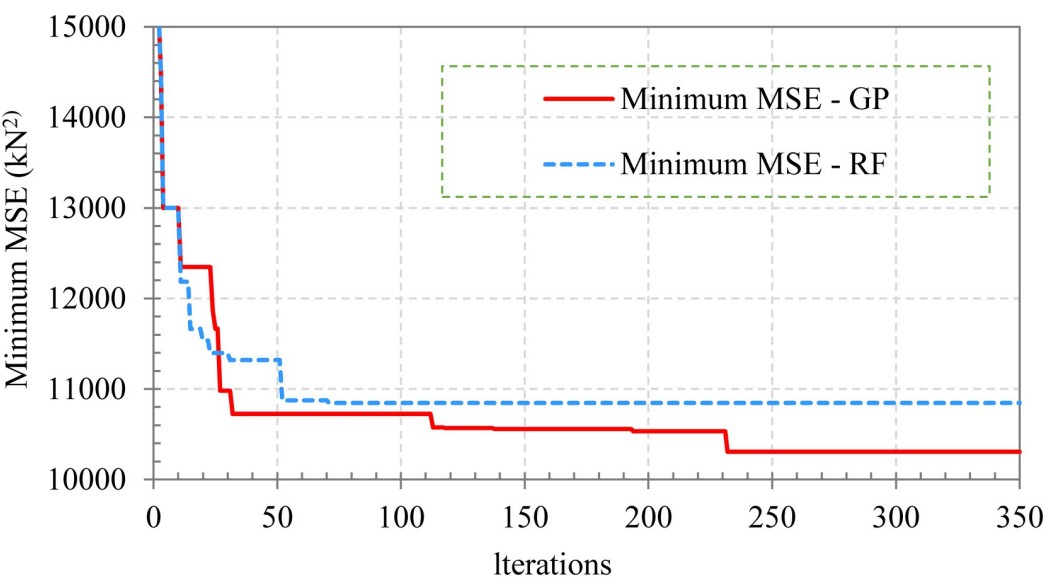

**Fig 3. Optimization process of BO algorithm.**

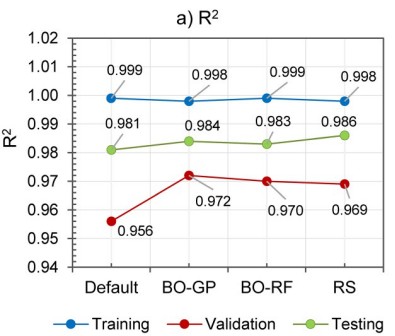
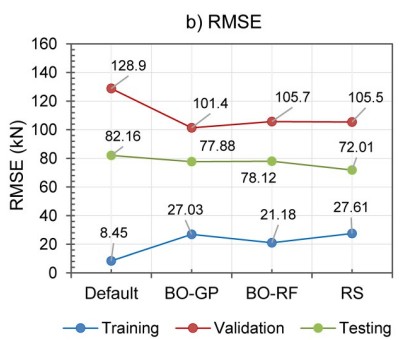
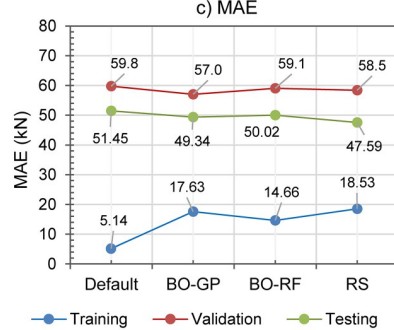

**Fig 4. Comparison of the learning results of the XGBoost model between the default parameters and the optimal parameters of the BO-GP, BO-RF and RS models according to the criteria: (a)—R2; (b)—RMSE; (c)–MAE.**

the XGBoost model are illustrated in Fig 4 (S2 Data). The results indicate that hyperparameter optimization has a significant impact on the training, validation, and testing performance of the XGBoost model.

All three optimization methods provide better training performance than the model with default hyperparameters, as demonstrated by the $R^2$, RMSE, and MAE metrics on the training, validation, and testing sets. The most pronounced changes are observed in the $R^2$, RMSE, and MAE values, on the training, validation, and testing sets, specifically with validated $R^2$ values of 0.956/0.972/0.970/0.969 for the default and optimized parameters by BO-GP, BO-RF, and RS. From the results obtained on the three optimization methods, it is evident that the BO-GP method is the method that gives higher model performance than the other two methods.

And it should be noted that the difference in results between the model with default hyperparameters and the optimized model is negligible on the training set, and sometimes the default hyperparameters even yield results. slightly better than the optimized model. This highlights the powerful learning capabilities of the XGBoost model and its ability to fine-tune hyperparameters to prevent overfitting when good training results are achieved.

In addition, the training results of the XGBoost model for the wall cross-section types are presented in Fig 5. Fig 5, presents a scatter plot with a regression line that visually compares the shear strength of reinforced concrete walls predicted by the optimized parameters of the XGBoost model using the Gaussian Process method. Specifically, Fig 5a shows the regression results for all cross-section types. Fig 5b–5d show the regression results for Rectangular, Barbell and Flange cross-sections, respectively. The results show that most of the regression points are close to the reference line, demonstrating the excellent performance of the XGB model. In addition, the regression results of the model for Bar-shaped wall cross-sections appear to be more accurate than other cross-section types. Specifically, the correlation coefficients on the training and test sets were $R^2 = 0.999$ and 0.975, respectively, while the Root Mean Square Error RMSE = 10.891 kN and 93.604 kN.

## 5. The importance of variables to the model's predictive ability

To evaluate the importance of input variables on the model's predictive ability, the SHAP (Shapley Additive Interpretation) [44] values technique was used. SHAP values are a technique for interpreting the output of machine learning models. It uses a game theoretic approach to measure each player's contribution to the outcome. The XGBoost model optimized according to the BO-GP solution is used as the main model to analyze the influence of input variables according to SHAP theory. The absolute SHARP value can be used to determine the influence

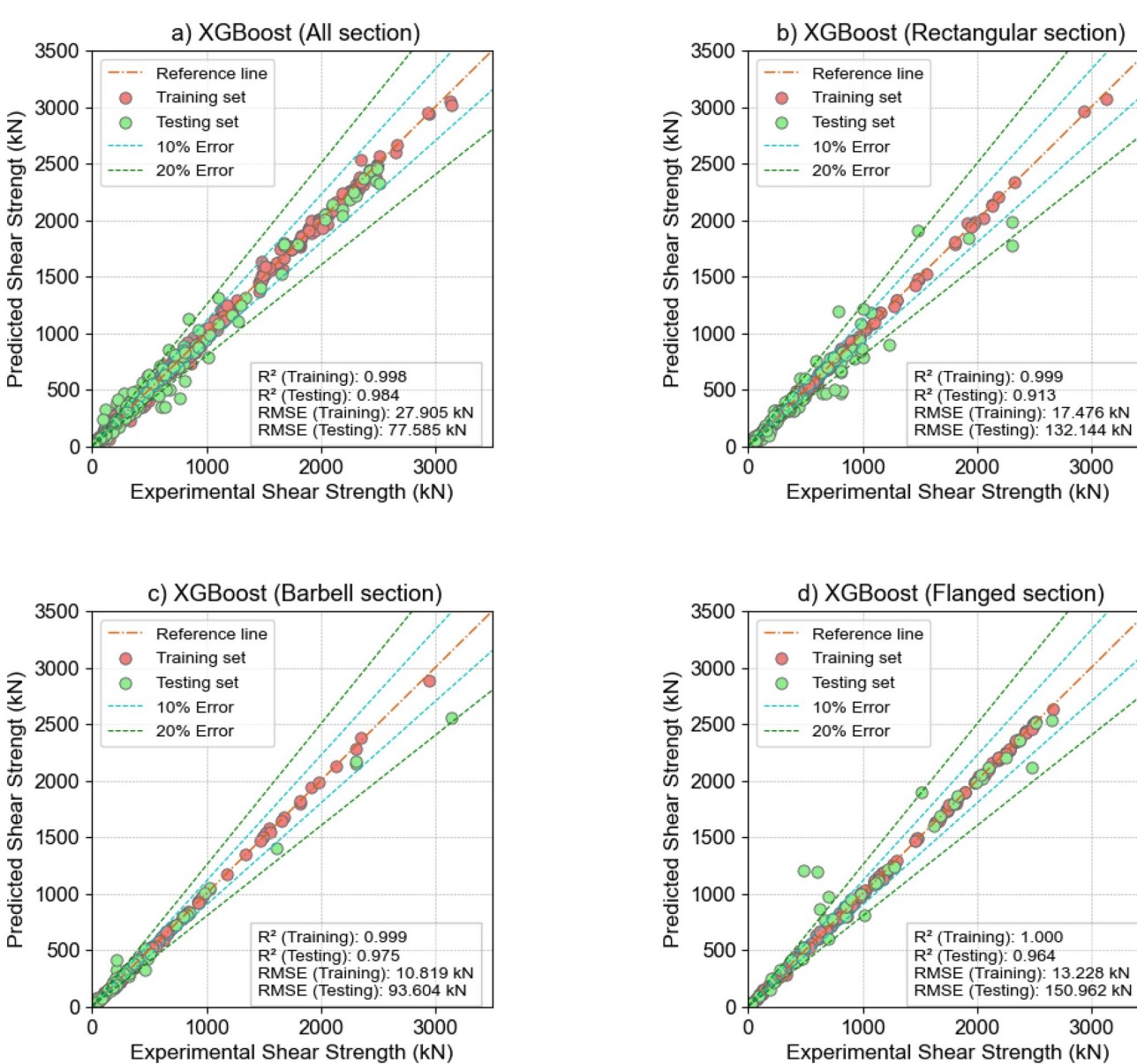

**Fig 5. The shear strength prediction results of the XGBoost model optimized by the Gaussian Process method for the following types of cross-sections: (a)—all section; (b)—Rectangular section; (c)—Barbell section; (d)—Flanged section.**

of each input characteristic on the model output value. Fig 6 illustrates the results of the SHAP analysis and provides valuable insights into the influence of each input variable characteristic in forecasting the shear strength of RC walls.

Based on the results, it can be inferred that the flange length ($X_5$) and wall length ($X_2$) are the most important characteristics affecting the shear strength of reinforced concrete walls. More specifically, when the flanged length value ($X_5$) increases to the maximum value of this variable (redpoint), the corresponding Shap value increases in the positive direction to more than 500. This shows that the shear strength of the wall increases significantly in proportion to the flanged length. Meanwhile, when the wall length value ($X_2$) increases, the maximum Shap value of this variable reaches about 1200, showing that the impact of this variable on the shear

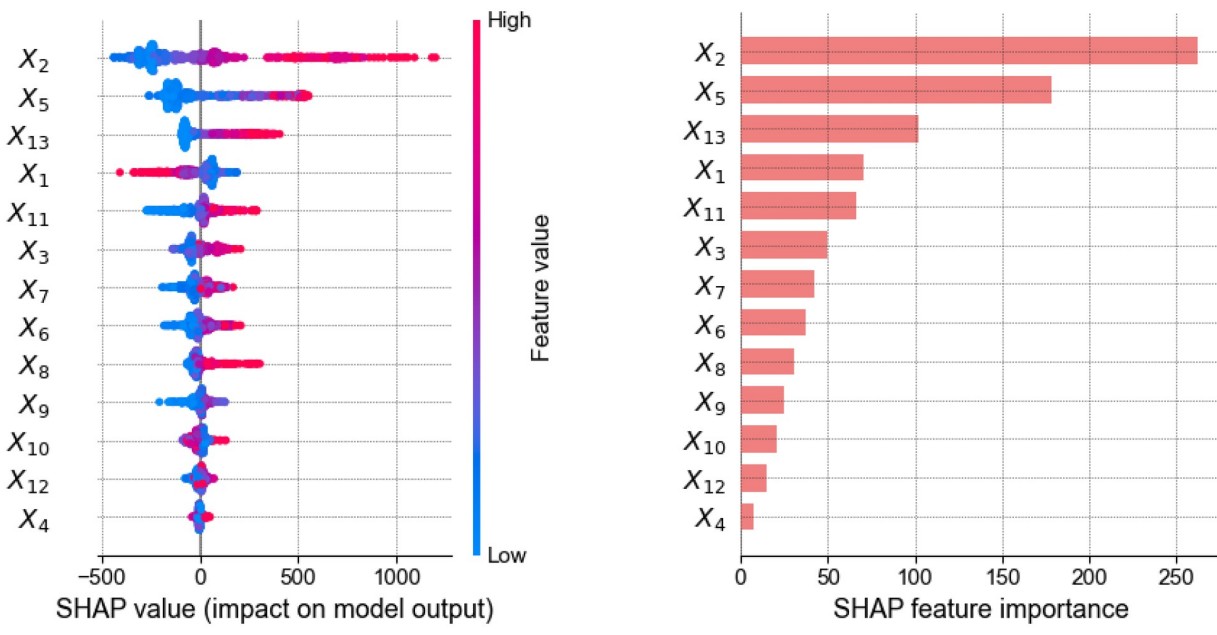

**Fig 6. SHAP summary plot and the relative importance of each feature of the XGBoost model.**

strength of the wall is even greater, that is, the longer the wall length, the higher the shear strength value. Additionally, the axial load ($X_{13}$) is also significant for the shear strength of RC walls, but it has been overlooked in the prediction equations by ACI 318–19 (ACI 2019) and Wood (1990).

Interestingly, the influence of longitudinal reinforcement ($X_{11}$) is greater than that of horizontal and vertical reinforcement ($X_9$ and $X_7$) in predicting shear strength. Notably, horizontal reinforcement primarily acts as ties, enhancing cohesion and preventing instability of the vertical bars. However, based on the results of the analysis in this study, other features, such as the vertical reinforcement content and the intrinsic shear capacity of the concrete, show a more significant contribution than the horizontal reinforcement in forming the shear resistance of RC walls. This observation does not diminish the essential role of transverse reinforcement in RC walls but highlights its relative influence compared to other factors, in this data set.

## 6. Compare the performance of the XGBoost model with current design codes

To evaluate the performance of the ML model, three semi-empirical shear strength determination models based on mechanical theory are used for comparison to evaluate the prediction accuracy of the XGBoost model. These are the models provided in ACI 318–19 (Chapter 11) (ACI 2019) [45, p. 14], ASCE/SEI 43–05 (ASCE 2005) [46], and by Wood (1990) [47], is given as follows:

- **ACI 318–19:**

$$V_n = \left( \alpha_c \lambda \sqrt{f_{ck}} + \rho_h f_{yh} \right) A_{cv} \leq 0.83 \sqrt{f_{ck}} A_{cw} \tag{5}$$

Where: $\alpha_c$ is the aspect ratio coefficient, $\alpha_c = 0.25$ when $h_w/l_w \leq 1.5$; $\alpha_c = 0.17$ when $h_w/l_w \geq 2.0$ and changes linearly between 0.25 and 0.17 for $h_w/l_w$ between 1.5 and 2.0 (wall length–$l_w$,

high wall–$h_w$); λ is a coefficient of variation that reflects the properties of concrete and is equal to 1.0 for normal strength concrete; $A_{cv}$ is the total area of the concrete limited by the thickness of the web and the length of the section in the direction of the considered shear force; and $A_{cw}$ is the total cross-sectional area of the wall;

- **ASCE/SEI 43–05:**

$$V_n = v_n d t_w \tag{6}$$

$$v_n = 0.69\sqrt{f_{ck}} - 0.28\sqrt{f_{ck}}\left(\frac{h_w}{l_w} - 0.5\right) + \frac{P}{4l_w t_w} + \rho_{se} f_{yh} \leq 1.66\sqrt{f_{ck}} \tag{7}$$

$$\rho_{se} = A\rho_v + B\rho_h \tag{8}$$

Where d = $0.6l_w$; $\rho_{se}$ is the equivalent reinforcing ratio combining $\rho_h$ and $\rho_v$ with coefficients A = 1; B = 0 for $h_w/l_w \leq 0.5$; A = $- h_w/l_w + 1.5$; B = $h_w/l_w - 0.5$ for $0.5 \leq h_w/l_w \leq 1.5$; and A = 0; B = 1 for $h_w/l_w \geq 1.5$.

- **Wood (1990):**

$$0.5\sqrt{f_{ck}}A_{cv} \leq V_n = \frac{A_{vf}f_{yv}}{4} \leq 0.83\sqrt{f_{ck}}A_{cv} \tag{9}$$

Where $A_{vf}$ is the total area of shear reinforcement, reinforcement is arranged along the height of the wall to improve shear strength; $f_{ck}$ is the compressive strength of concrete, $f_{yv}$—Strength of vertical reinforcement in the web, $A_{cw}$ is the total cross-sectional area of the wall.

The three standard models and the XGBoost model were used to predict the shear strength of walls for a dataset consisting of 1,057 samples, which included three different cross-section types and a larger height-to-width aspect ratio of 1.5. The comparison results between the XGBoost model and the standard models, as well as the practice code reference models, are presented in and Fig 7 (S3 Data).

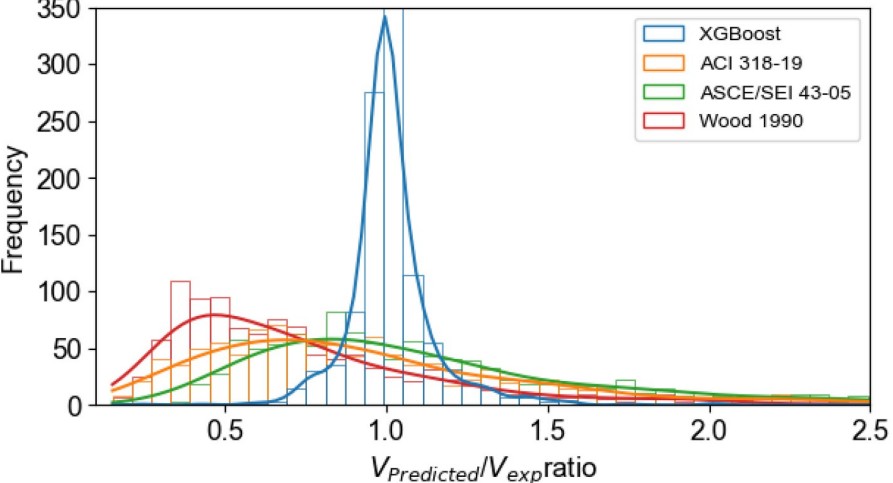

**Fig 7. Results of predicting shear resistance using mechanical model and XGBoost model.**

**Table 3. Performance comparison between the ML model and the experimental and predictive scale-mechanical model.**

| Models | Predicted to experiment ratio ($V_{predicted}/V_{exp}$) | | | | |
|---|---|---|---|---|---|
| | Min. | Max. | Mean | St.D. | COV (%) |
| ACI 318–19 | 0.151 | 4.742 | 0.980 | 0.614 | 62.64 |
| ASCE/SEI 43–05 | 0.325 | 4.953 | 1.173 | 0.647 | 55.19 |
| Wood 1990 | 0.166 | 4.830 | 0.764 | 0.506 | 66.15 |
| XGBoost model | 0.244 | 2.022 | 1.013 | 0.134 | 13.27 |

Comparisons are shown between the predicted shear strength ratios and the experimentally obtained shear strengths for the samples in the database. The model evaluations include the mean standard deviation (St.D), mean value (Mean), maximum value (Max.), minimum value (Min.), and coefficient of variation (COV%) of the ratio between the predictions and the experimental results.

The results from Fig 7 and Table 3 indicate that the prediction ratio of shear strength for walls between models and experiments, according to current standard models, shows high average errors and large variability. The XGBoost model is optimized according to the BO-GP solution for excellent performance with a high Predicted-to-experiment ratio and lowest standard deviation. That demonstrates the accuracy and stability of the XGBoost model in predicting the shear strength of RC walls.

Furthermore, the analytical results comparing the predicted wall shear strength ratio with the experimental values on different specimens, shown in Fig 8 (S4 Data), show the mean predicted value and the standard deviation (St.D) prediction range of the model. The analysis shows that walls with lower $h_w/l_w$ ratios ($\leq 1.0$) exhibit greater variation in prediction accuracy than walls with higher $h_w/l_w$ ratios (ranging from 1.0 to 3.5), indicating less consistency in predictions for smaller aspect ratios. This variation may be the result of complex interactions associated with lower $h_w/l_w$ ratios, which are less effectively captured by the model, while higher ratios simplify predictions due to more uniform geometric and structural features. These findings emphasize the importance of taking $h_w/l_w$ ratio into account in shear wall design to improve prediction reliability and ensure safety. Notably, the XGBoost model outperforms conventional semi-empirical models, demonstrating superior predictive performance on both low aspect ratio walls with hw/lw $\leq 1.5$ and high aspect ratio walls with hw/lw $> 1.5$. The robustness and flexibility of the model make it an effective tool for predicting shear strength in a variety of wall configurations.

## 7. Conclusions

The study utilized the XGBoost model to analyze a dataset comprising 1057 RC wall samples with various cross-section types and aspect ratios (ratios $>1.5$, $= 1.5$, $<1.5$). Key findings from this research are summarized as follows:

The optimization of the XGBoost model using Bayesian GP, RF, and RS methods demonstrated the importance of hyperparameter tuning compared to default hyperparameters. All three hyperparameter optimization models significantly improved performance over the default model, with the GP method providing the best results. The XGBoost model, optimized using the BO-GP method, achieved stable prediction performance across all cross-section types and the combined dataset of the three cross-section types, with $R^2$ scores on the test set of 0.984/0.913/0.975/0.964 for all sections, Rectangular sections, Barbell sections, and Flanged sections, respectively.

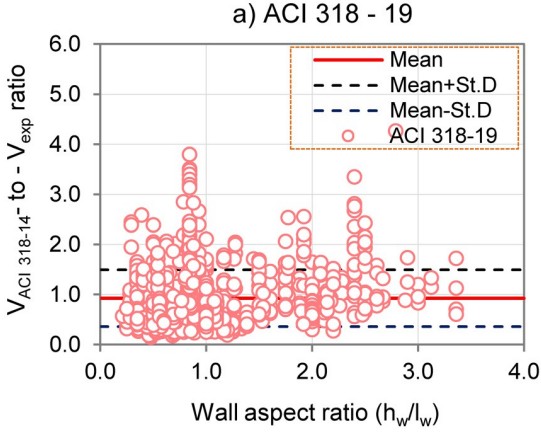
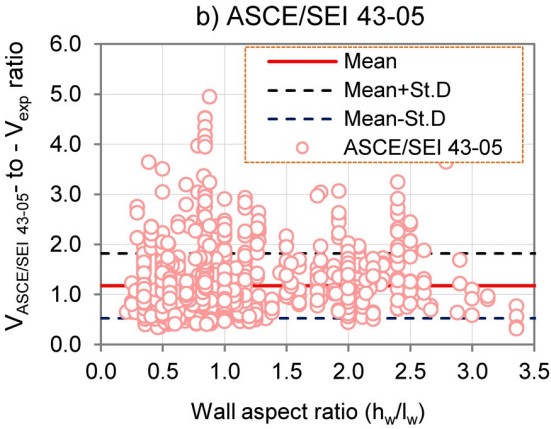
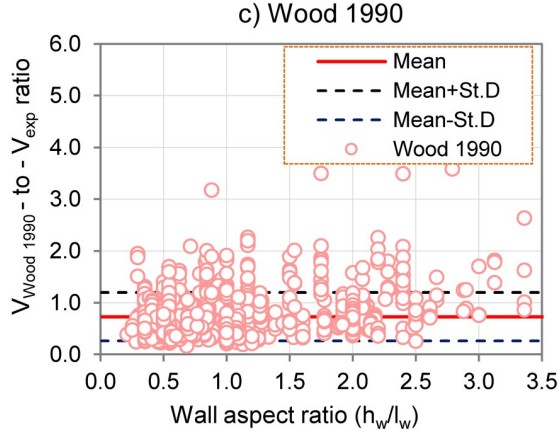
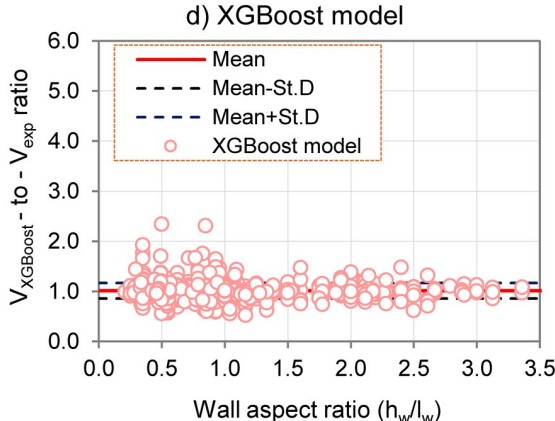

**Fig 8. Shear strength predicted by mechanics—Based models: a) ACI 318–19; b) ASCE/SEI 43–05; c) Wood 1990; and d) XGBoost model according to different aspect ratios ($h_w/l_w$ ratios >1.5, = 1.5, <1.5).**

The SHAP value technique is used to explain the predictive ability of the XGBoost model and analyze the role of input variables for all types of RC wall cross-sections. The results show that the two input factors, flange length ($b_f$) and wall length ($l_w$), are the most important characteristics affecting the shear strength of RC walls, for this dataset.

The optimized XGBoost model was also compared with existing standards and codes. The results demonstrated that the XGBoost model significantly improved the predictive performance compared with traditional design standards such as ACI 318–19, ASCE/SEI 43–05, and Wood 1990. Furthermore, the study results showed that the XGBoost model was capable of effectively predicting shear strength within the range of aspect ratios $h_w/l_w$ >1.5. These findings highlight the robustness of the XGBoost model in accurately predicting the shear strength of reinforced concrete walls, emphasizing the advantages of advanced machine learning techniques over traditional design methods. However, it should be noted that the XGBoost model does not have extrapolation capabilities, so the model's accuracy is only guaranteed within the range of input variable values it was trained on. This can be improved by using a more general training dataset and using machine learning models that are not limited in extrapolation capabilities.

## Supporting information

**S1 Data. Data from Fig 3.**
(CSV)

**S2 Data. Data from Fig 4.**
(CSV)

**S3 Data. Data from Fig 7.**
(CSV)

**S4 Data. Data from Fig 8.**
(CSV)

**S1 Table. Example of data used in Table 1.**
(CSV)

## Author Contributions

**Conceptualization:** Hoa Thi Trinh, Tuan Anh Pham.

**Investigation:** Duy Hung Nguyen.

**Supervision:** Vu Dinh Tho.

**Visualization:** Vu Dinh Tho, Duy Hung Nguyen.

**Writing – original draft:** Hoa Thi Trinh.

**Writing – review & editing:** Tuan Anh Pham.

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
