## [Decision Letter · Decision Letter 0]

14 Aug 2024

PONE-D-24-29804The research explores the predictive capacity of the shear strength of reinforced concrete walls with different cross-sectional shapes using the XGBoost modelPLOS ONE

Dear Dr. Pham,

Thank you for submitting your manuscript to PLOS ONE. After careful consideration, we feel that it has merit but does not fully meet PLOS ONE’s publication criteria as it currently stands. Therefore, we invite you to submit a revised version of the manuscript that addresses the points raised during the review process.

We look forward to receiving your revised manuscript.

Kind regards,

Afaq Ahmad, PhD

Academic Editor

PLOS ONE

Journal Requirements:

4. We note that your Data Availability Statement is currently as follows: All relevant data are within the manuscript and its Supporting Information files

Additional Editor Comments:

Author needs to provide the description of some critical parts of the study.

1- Abstract could be more informative by providing results. I prefer to see some results in the abstract.

2- The introduction needs to be more emphasized on the research work with a detailed explanation of the whole process considering past, present and future scope. How the present study gives more accurate results than previous studies? It needs to be strengthened in terms of recent research in this area with possible research gaps. It is strongly recommended to add a recent literature.

3- Please avoid the basic details about the methodology in the introduction portion, the introduction portion, please use only the latest reference. Please reduce these sections.

4- Please describe the important and novelty of the selected problem, data details. Please provide details about the selected problem. Please include the validation process on the unique problem.

5- In the conclusion section, the limitations of this study, suggested improvements of this work, and future directions should be added

The author needs to address the abovementioned points for the betterment of the manuscript.

Reviewers' comments:

Reviewer's Responses to Questions

**Comments to the Author**

1. Is the manuscript technically sound, and do the data support the conclusions?

Reviewer #1: No

Reviewer #2: Yes

2. Has the statistical analysis been performed appropriately and rigorously? 

Reviewer #1: Yes

Reviewer #2: Yes

3. Have the authors made all data underlying the findings in their manuscript fully available?

Reviewer #1: No

Reviewer #2: No

4. Is the manuscript presented in an intelligible fashion and written in standard English?

Reviewer #1: Yes

Reviewer #2: Yes

5. Review Comments to the Author

Reviewer #1: The authors presents a well-written and structured manuscript on the shear strength estimation of reinforced concrete walls. Specifically, In this study, machine learning techniques such as XGBoost models are trained and the optimal among them is proposed to predict the shear strength of reinforced concrete walls with different cross-sectional shapes. The conclusions are well-supported by the quantity and quality of the data and results. However, some minor comments and suggestions are provided to aid in further enhancing the manuscript's overall quality.

I. The abstract should be concise, focusing on the methodology employed and the key findings. A clear and succinct summary will enhance the paper's readability and impact.

II. Consider adding a section after the introduction titled "Research Significance" to emphasize the need for further research on this subject and highlight the novelty and innovation of your work. This will provide context and motivation for your study.

III.

IV. In order to assess the reliability of soft computing models, researchers often use statistical indices such as MSE. The authors are encouraged to include a recently proposed performance index, the a20 index (see the last equation in Table 8 of https://doi.org/10.1016/j.ultras.2024.107347 and eq. 11 in s00521-021-06004-8).

V. To bolster their results, the authors are urged to include in the database used for model training both the experimental ‘true’ values and those predicted by the optimal proposed model, along with corresponding publication details.

VI. Transitions between sections should be smoother to enhance readability.

VII. The conclusion section should be modified and improved.

VIII. The literature review presented in the manuscript is not sufficiently comprehensive. The authors are encouraged to refer to extensive state-of-the-art reports in soft computing techniques such as ‘Predicting the shear strength of reinforced concrete beams using Artificial Neural Networks,’ ‘Convolution-based ensemble learning algorithms to estimate the bond strength of the corroded reinforced concrete,’ ‘Predicting the thermal conductivity of soils using integrated approach of ANN and PSO with adaptive and time-varying acceleration coefficients,’ ‘Analysis and prediction of the effect of Nanosilica on the compressive strength of concrete with different mix proportions and specimen sizes using various numerical approaches,’ ‘Predicting the unconfined compressive strength of granite using only two non-destructive test indexes,’ ‘Genetic prediction of ICU hospitalization and mortality in COVID‐19 patients using artificial neural networks,’ and ‘Developing bearing capacity model for geogrid-reinforced stone columns improved soft clay utilizing MARS-EBS hybrid method.’. Detailed and in depth state-of-the-art report can be found in https://doi.org/10.1016/j.jobe.2023.108369

IX. How do you address the potential overfitting of the hybrid model? A short paragraph about this crucial issue will also add more value in their work.

X. Adding a short section on the limitations of the proposed models, titled 'Limitations and Future Work,' will enhance the value of the submitted work.

XI. A thorough proofreading is essential to address typos and language errors. Improving the manuscript's English usage will enhance its overall readability and professionalism.

Addressing these specific comments will significantly enhance the quality and suitability of your manuscript for publication in an international journal.

Reviewer #2: please

1- add correlation matrix of inputs

2- add Ref for equation 2 and 3

3- add criteria for stopping optimization

4- add some ref to support the results of the section 4:

The analysis results show clearly, the flange length (X5) and wall length (X2) has the

greatest influence on the shear strength value of RC walls. This proves that, in addition to

the wall body, the flange also significantly participates in the wall's shear capacity.

- cite recent related paper like what follows to enrich introduction and compare your paper with them and the gap you cover:

"Ensemble techniques and hybrid intelligence algorithms for shear strength prediction of squat reinforced concrete walls." 1, 8(1), 37. https://doi.org/10.12989/acd.2023.8.1.039

"Failure mode detection of reinforced concrete shear walls using ensemble deep neural networks." International Journal of Concrete Structures and Materials 16, no. 1 (2022): 33. https://doi.org/10.1186/s40069-022-00522-y

Response estimation of reinforced concrete shear walls using artificial neural network and simulated annealing algorithm. In Structures (Vol. 34, pp. 1155-1168). Elsevier.https://www.sciencedirect.com/science/article/pii/S2352012421007657

6. PLOS authors have the option to publish the peer review history of their article (what does this mean?). If published, this will include your full peer review and any attached files.

Reviewer #1: No

Reviewer #2: No

---

## [Author Response · Author response to Decision Letter 0]

25 Sep 2024

RESPONSES OF THE REVIEWER’S COMMENTS

Dear Editor and reviewers, we thank for your nice and constructive comments, which help us to improve the quality of our work. 

Responses to Editor:

1- Abstract could be more informative by providing results. I prefer to see some results in the abstract.

Respond: We have added additional results information to the Abstract section, highlighted in red and presented as follows:

The results show that Gaussian Process emerged as the most efficient solution compared to other optimization algorithms, providing the lowest Mean Square Error and achieving a prediction R2 of 0.998 for the training set, 0.972 for the validation set and 0.984 for the test set, while BO - Random Forest and Random Search performed as well on the training and test sets as Gaussian Process but significantly worse on the validation set, specifically R2 on the validation set of BO - Random Forest and Random Search were 0.970 and 0.969 respectively over the entire dataset including all cross-sectional shapes of the RC wall.

2- The introduction needs to be more emphasized on the research work with a detailed explanation of the whole process considering past, present and future scope. How the present study gives more accurate results than previous studies? It needs to be strengthened in terms of recent research in this area with possible research gaps. It is strongly recommended to add a recent literature.

Respond: We have added analyses related to the limitations of existing studies, research gaps, and some recently published literature, specifically highlighted in red in the introduction section:

3- Please avoid the basic details about the methodology in the introduction portion, the introduction portion, please use only the latest reference. Please reduce these sections.

Respond: We found the Abstract section to be quite concise and to highlight the main results. If this section were reduced, we fear that the reader would not find the research results interesting.

4- Please describe the important and novelty of the selected problem, data details. Please provide details about the selected problem. Please include the validation process on the unique problem.

Respond: We have added section 2. Research significance, the content highlighted in red in the revised manuscript as follows:

2. Research significance

To address the limitations of previous studies, this paper proposes the following approach to apply the ML model in predicting the shear strength of reinforced concrete walls:

1) The large dataset of RC walls collected and processed includes 1057 samples with three different cross-sectional shapes.

2) A detailed study was performed on the XGBoost model, with the parameter sets for the XGBoost model determined through three different optimization methods.

3) The role of input variables is evaluated using SHAP values for the XGBoost model, providing an explanation of the model's predictive ability.

4) The prediction ability of the XGBoost model is compared with standard design codes and existing benchmark models.

By implementing this method, the study is expected to overcome the limitations of previous studies and provide a more effective method to evaluate the shear capacity of RC walls.

5- In the conclusion section, the limitations of this study, suggested improvements of this work, and future directions should be added

The author needs to address the abovementioned points for the betterment of the manuscript.

Respond: We thank to editor’s comments. We have added limitations and solutions at the end of the conclusion section, as follows:

However, it should be noted that the XGboost model does not have extrapolation capabilities, so the model's accuracy is only guaranteed within the range of values it was trained on. This can be improved by using a more general training dataset and using machine learning models that are not limited in extrapolation capabilities.

Responses to Reviewer 1 

Dear reviewer #1, we thank for your nice and constructive comments, which help us to improve the quality of our work. Especially comments on addressing the overfitting problem and limitations of the study.

1. Comment 1

The abstract should be concise, focusing on the methodology employed and the key findings. A clear and succinct summary will enhance the paper's readability and impact.

Respond: This is useful advice, we have tried to rewrite the abstract to clarify the research content and contributions in it.

2. Comment 2

Consider adding a section after the introduction titled "Research Significance" to emphasize the need for further research on this subject and highlight the novelty and innovation of your work. This will provide context and motivation for your study. 

Respond: We appreciate this comment, it makes the manuscript much clearer. We have added section 2.

2. Research significance

To address the limitations of previous studies, this paper proposes the following approach to apply the ML model in predicting the shear strength of reinforced concrete walls:

1) The large dataset of RC walls collected and processed includes 1057 samples with three different cross-sectional shapes.

2) A detailed study was performed on the XGBoost model, with the parameter sets for the XGBoost model determined through three different optimization methods.

3) The role of input variables is evaluated using SHAP values for the XGBoost model, providing an explanation of the model's predictive ability.

4) The prediction ability of the XGBoost model is compared with standard design codes and existing benchmark models.

By implementing this method, the study is expected to overcome the limitations of previous studies and provide a more effective method to evaluate the shear capacity of RC walls.

3. Comment 3

In order to assess the reliability of soft computing models, researchers often use statistical indices such as MSE. The authors are encouraged to include a recently proposed performance index, the a20 index (see the last equation in Table 8 of https://doi.org/10.1016/j.ultras.2024.107347 and eq. 11 in s00521-021-06004-8).

Respond: We thank to reviewer’s comments. We found it necessary to add a performance indicator to make the results more objective. However, we have no experience working with the A20 indicator, so we decided to use the slightly more popular MAE (Mean Absolute Error) indicator.

Mean Absolute Error (MAE): MAE=1/N ∑_(j=1)^N▒| y_j-y_(t,j) | (4)

where yj is the actual shear strength of the jth sample in the dataset; y𝑡,𝑗 is the predicted shear strength of the jth sample obtained from the ML model; y ® is the mean value of the actual shear strength of the data set; N is the total number of samples in the dataset.

4. Comment 4

To bolster their results, the authors are urged to include in the database used for model training both the experimental ‘true’ values and those predicted by the optimal proposed model, along with corresponding publication details.

Respond: We will consider publishing the data in the article. The data collection method has been described in detail in section 3.1. However, any requests for data will be responded to by contacting Corresponding author.

5. Comment 5

Transitions between sections should be smoother to enhance readability.. 

Respond: We thank to reviewer’s comments. We have attempted to edit the content of the forwarding sections.

6. Comment 6

The conclusion section should be modified and improved.

Respond: We thank to reviewer’s comments. We found it very helpful. The conclusion has been rewritten as recommended.

7. Comment 7

The literature review presented in the manuscript is not sufficiently comprehensive. The authors are encouraged to refer to extensive state-of-the-art reports in soft computing techniques such as ‘Predicting the shear strength of reinforced concrete beams using Artificial Neural Networks,’ ‘Convolution-based ensemble learning algorithms to estimate the bond strength of the corroded reinforced concrete,’ ‘Predicting the thermal conductivity of soils using integrated approach of ANN and PSO with adaptive and time-varying acceleration coefficients,’ ‘Analysis and prediction of the effect of Nanosilica on the compressive strength of concrete with different mix proportions and specimen sizes using various numerical approaches,’ ‘Predicting the unconfined compressive strength of granite using only two non-destructive test indexes,’ ‘Genetic prediction of ICU hospitalization and mortality in COVID‐19 patients using artificial neural networks,’ and ‘Developing bearing capacity model for geogrid-reinforced stone columns improved soft clay utilizing MARS-EBS hybrid method.’. Detailed and in depth state-of-the-art report can be found in https://doi.org/10.1016/j.jobe.2023.108369

Respond: We thank to reviewer’s comments. We found these studies useful, so we have chosen to cite two relevant studies in the revised manuscript:

Recently, Machine Learning (ML) based models have demonstrated their effectiveness in forecasting the shear strength of various structural elements such as beams [11], [12], [13,14]

[13]. L. Cavaleri, M.S. Barkhordari, C.C. Repapis, D.J. Armaghani, D.V. Ulrikh, P.G. Asteris, Convolution-based ensemble learning algorithms to estimate the bond strength of the corroded reinforced concrete, Construction and Building Materials 359 (2022) 129504.

[14]. P.G. Asteris, D.J. Armaghani, G.D. Hatzigeorgiou, C.G. Karayannis, K. Pilakoutas, Predicting the shear strength of reinforced concrete beams using Artificial Neural Networks, Computers and Concrete, An International Journal 24 (2019) 469–488.

8. Comment 8

How do you address the potential overfitting of the hybrid model? A short paragraph about this crucial issue will also add more value in their work. 

Respond: We found this very useful and have added content about solutions to avoid overfitting in section 4.1 of the revised manuscript.

It is important to note that to avoid overfitting during training and optimization, two techniques have been applied: (1) Subsample and (2) K-fold CV. In which, the Subsample technique uses a certain proportion of input variables during training, which helps create simpler trees and avoid overfitting. Meanwhile, the K-Fold technique is used on the training data set itself, allowing the model to be trained/validated during the optimization process, all single data fold in the training set is in turn fed into training/validation, leading to training results that avoid overfitting.

9. Comment 9

Adding a short section on the limitations of the proposed models, titled 'Limitations and Future Work,' will enhance the value of the submitted work.". 

Respond: We thank to reviewer’s comments. We found it helpful But we found that adding a short paragraph was really unnecessary, so we added a small paragraph about the study's limitations and solutions at the end of the conclusion as follows:

However, it should be noted that the XGboost model does not have extrapolation capabilities, so the model's accuracy is only guaranteed within the range of input variable values it was trained on. This can be improved by using a more general training dataset and using machine learning models that are not limited in extrapolation capabilities.

10. Comment 10

A thorough proofreading is essential to address typos and language errors. Improving the manuscript's English usage will enhance its overall readability and professionalism.

Respond: We appreciate the reviewers' comments. We have tried our best to correct the manuscript.

Responses to Reviewer 2 

Dear reviewer #1, we thank for your nice and constructive comments, which help us to improve the quality of our work. 

1. Comment 1

Add correlation matrix of inputs. 

Respond: We thank to reviewer’s comments. We found it very helpful and had added correlation matrix of input in Figure 2 in the revised manuscript and also adds some analysis of input variable correlations :

Figure 2. Correlation matrix of the features with data 1057samples 

Figure 2 shows the correlation matrix of the data set, which includes 13 input variables and 1 output variable. The matrix displays the correlation coefficients between each pair of variables, where a correlation value of 1 represents a perfect positive correlation, -1 represents a perfect negative correlation, and 0 represents no correlation. The correlation matrix helps us understand the relationship between different variables and how they relate to each other. Initial analysis shows that there are both positive and negative correlations between variables and that pairs of highly correlated attributes are more interdependent. Specifically, the highest correlation coefficient is 0.89 between the two variables X8 and X10, demonstrating a close relationship between these characteristics. Additionally, geometrical parameters and loads applied to the wall have the highest correlation with output performance. Understanding the correlation matrix can help determine which features are important to the resulting characterization and which features are redundant, useful for further analysis and modeling.

2. Comment 2

Add Ref for equation 2 and 3. 

Respond: We agreed with reviewer’s comments. We found it right to had added reference for eqution 2, 3 as follow:

To evaluate the performance of the established models, statistical parameters, including Correlation coefficient (R2) [41], Root Mean Square Error (RMSE) [42], and mean absolute error (MAE) [42],

3. Comment 3

Add criteria for stopping optimization 

Respond: We found reviewer’s comments is very helpful. We have added the optimal stopping condition into the manuscript as follows (line 222-223):

 The optimization process will stop after the algorithm has performed at least 100 iterations, without the optimal result changing.

4. Comment 4

Add some ref to support the results of the section 4:

The analysis results show clearly, the flange length (X5) and wall length (X2) has the

greatest influence on the shear strength value of RC walls. This proves that, in addition to

the wall body, the flange also significantly participates in the wall's shear capacity.

Respond: We thank to reviewer’s comments. We found the input variable impact analysis part to be a bit lacking, so we rewrote it as follows:

Based on the results, it can be inferred that the flange length (X5) and wall length (X2) are the most important characteristics affecting the shear strength of reinforced concrete walls. More specifically, when the flanged length value (X5) increases to the maximum value of this variable (redpoint), the corresponding Shap value increases in the positive direction to more than 500. This shows that the shear strength of the wall increases significantly in proportion to the flanged length. Meanwhile, when the wall length value (X2) increases, the maximum Shap value of this variable reaches about 1200, showing that the impact of this variable on the shear strength of the wall is even greater, that is, the longer the wall length, the higher the shear strength value.

5. Comment 15

cite recent related paper like what follows to enrich introduction and compare your paper with them and the gap you cover:

"Ensemble techniques and hybrid intelligence algorithms for shear strength prediction of squat reinforced concrete walls." 1, 8(1), 37. https://doi.org/10.12989/acd.2023.8.1.039

"Failure mode detection of reinforced concrete shear walls using ensemble deep neural networks." International Journal of Concrete Structures and Materials 16, no. 1 (2022): 33. https://doi.org/10.1186/s40069-022-00522-y

Response estimation of reinforced concrete shear walls using artificial neural network and simulated annealing algorithm. In Structures (Vol. 34, pp. 1155-1168). Elsevier.https://www.sciencedirect.com/science/article/pii/S2352012421007657

Respond: We thank to reviewer’s comments. We found these studies to be very meaningful, so we have cited two studies in the Introduction, namely:

[19] M.S. Barkhordari, L.M. Massone, Failure mode detection of reinforced concrete shear walls using ensemble deep neural networks, International Journal of Concrete Structures and Materials 16 (2022) 33.

[25] M.S. Barkhordari, L.M. Massone, Ensemble techniques 

---

## [Editor Report · Decision Letter 1]

9 Oct 2024

The research explores the predictive capacity of the shear strength of reinforced concrete walls with different cross-sectional shapes using the XGBoost model

PONE-D-24-29804R1

Dear Dr. Pham,

We’re pleased to inform you that your manuscript has been judged scientifically suitable for publication and will be formally accepted for publication once it meets all outstanding technical requirements.

Kind regards,

Afaq Ahmad, PhD

Academic Editor

PLOS ONE

Additional Editor Comments (optional):

The authors have adeptly addressed the constructive comments and suggestions provided during the review process, showcasing their commitment to enhancing the quality and rigor of their manuscript. Their thoughtful revisions and meticulous attention to detail have resulted in a strengthened and more robust final version of the paper. As such, it is recommended that the manuscript be accepted for publication, as it contributes significantly to the existing body of knowledge in the field and reflects the dedication of the authors to scholarly excellence.
---

## [Editor Report · Acceptance letter]

14 Oct 2024

PONE-D-24-29804R1 

PLOS ONE

Dear Dr. Pham, 

I'm pleased to inform you that your manuscript has been deemed suitable for publication in PLOS ONE. Congratulations! Your manuscript is now being handed over to our production team.

Kind regards, 

on behalf of

Dr Afaq Ahmad 

Academic Editor

PLOS ONE